# Predictors of shared decision-making among treatment-seeking emerging adults in primary care and community addiction and mental health settings: A cross-sectional study

Tyler Marshall[1], Karin Olson[2], Adam Abba-Aji[1], Xin-Min Li[1], Richard Lewanczuk[3], Sunita Vohra[1,4]*

1 Department of Psychiatry, Faculty of Medicine & Dentistry, College of Health Sciences, University of Alberta, Edmonton, Alberta, Canada, 2 Faculty of Nursing, University of Alberta, Edmonton, Alberta, Canada, 3 Department of Medicine, Faculty of Medicine & Dentistry, College of Health Sciences, University of Alberta, Edmonton, Alberta, Canada, 4 Department of Pediatrics, Faculty of Medicine & Dentistry, College of Health Sciences, University of Alberta, Edmonton, Alberta, Canada

* svohra@ualberta.ca

## Abstract

### Background

Shared decision-making (SDM) is a process in which healthcare providers (HCPs) and patients make health-related decisions collaboratively, guided by the best available evidence. Previous research suggests that emerging adults (aged 18–29) with mental health concerns might prefer SDM over traditional approaches; however, it remains unclear whether prevalent symptoms of anxiety, depression, or health-related quality of life (HRQL) are associated with the level of SDM that occurs during a clinical encounter.

### Objective

This study explored whether prevalent symptoms of anxiety, depression or HRQL among emerging adults were associated with the perceived level of SDM involvement during a single clinic visit at a primary care or community addiction and mental health (AMH) setting.

### Methods

A cross-sectional survey was conducted using a subset of data (emerging adults and their HCPs) obtained from an overarching study on SDM in adults (18–64 years) in Alberta, Canada. Sociodemographic data were collected and reported descriptively. SDM was the primary outcome variable and was measured dyadically (i.e., the mean score between HCPs and patients) using the Alberta Shared Decision-Making Instrument (ASK-MI). Symptoms of patient anxiety/depression and HRQL were measured using the Hospital Anxiety and Depression Scale (HADS) and the EQ-5D-5L.

**Data availability statement:** All relevant data are within the paper and its Supporting Information files.

**Funding:** This research was funded by the generous support of the Stollery Children's Hospital Foundation through the Women and Children's Health Research Institute and by the Faculty of Graduate Studies and Research (RES0033014) at the University of Alberta.. The funders had no role in study design, data collection and analysis, decision to publish, or preparation of the manuscript.

**Competing interests:** The authors have declared that no competing interests exist.

Pearson R correlation matrices were conducted to explore relationships between SDM, anxiety/depression, HRQL, and demographic variables.

## Results

Forty-two emerging adult patients and 31 HCP dyads were recruited from six community AMH settings and eight primary care settings. The mean SDM dyad rating was 8.69 (SD, ±2.01), indicating an "excellent" level of SDM. Symptoms of anxiety, depression, and HRQL were not significantly correlated with SDM dyad ratings during the clinic visit. Post hoc analyses showed that patient age was inversely related to SDM dyad ratings; $R = -0.34$, $p = 0.03$.

## Discussion

In this study, emerging adults reported high levels of perceived engagement in SDM, regardless of their HRQL or symptoms of anxiety and depression. However, several limitations, such as the risk of performance bias, should be considered when interpreting these findings. To strengthen the evidence base, future research should aim to address these limitations.

## Introduction

Emerging adults (individuals aged 18–29 years) have the highest rates of anxiety and depression among all adult age groups [1,2]. Early access to mental healthcare remains a critical public health issue, disproportionately impacting emerging adults and resulting in high costs and poor outcomes [3–5]. Moreover, emerging adults who successfully access professional mental health services are less likely to follow their treatment plans compared to older adults, which often leads to higher relapse rates and worse outcomes [4]. Evidence suggests that adopting more inclusive and developmentally suitable approaches to care—such as involving young adults in decisions about their mental health treatment—can improve engagement, satisfaction, and treatment outcomes [6].

Shared decision-making (SDM) is a clinical process in which patients or clients and their healthcare providers (HCPs) collaborate to make evidence-informed health decisions, such as setting health goals and choosing treatment options [7]. SDM emphasizes patients' values, preferences, and needs, which are essential for delivering evidence-informed and person-centred care [8,9]. We view SDM as occurring along a spectrum ranging from paternalism, where HCPs have complete control over health decisions, to full patient autonomy (or consumerism), with both the patient and provider guided by the best available scientific evidence.**Fig 1** displays our conceptual framework of SDM. Evidence from systematic reviews generally indicates positive outcomes associated with SDM [10–12]. Studies suggest SDM leads to patients being more likely to receive evidence-based treatments, and patient-reported outcomes, such as therapeutic alliance, treatment engagement, and adherence,

## Conceptual framework of shared decision-making

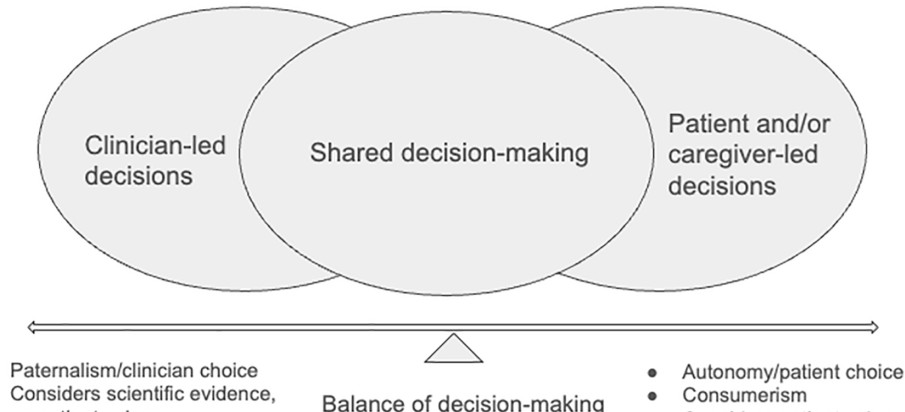

**Fig 1. Conceptual framework of shared decision-making.** SDM exists along a continuum that balances patient values, preferences, and autonomy with clinician expertise and the best available evidence. The level of SDM depends on the depth of dialogue, information exchange, and collaborative deliberation between patients and clinicians. At one end, characterized by paternalism, decisions are primarily clinician-driven, with minimal patient input into goals or treatment choices. At the other end, characterized by consumerism, decisions are largely patient-driven, with clinicians providing care based predominantly on patient preferences. Across the continuum, SDM aims to integrate patient priorities, clinical knowledge, and the best available scientific evidence, while patients or substitute decision-makers retain ultimate authority to consent to or refuse care.

may also enhance outcomes [10,11]. More recently, similar findings have been observed, indicating the relevance of SDM among individuals with mental health conditions such as anxiety and depression [10,12]. However, to our knowledge, limited empirical evidence exists regarding the use of SDM in specific age groups, such as emerging adults [10].

Hypothetically, SDM may be particularly suitable for emerging adults facing common mental health issues such as anxiety and depression. Evidence shows that emerging adults are more likely to delay seeking mental health treatment and prefer self-reliance compared to adolescents and older adults [13,14]. As a result, they may be at risk of self-medication with substances like nicotine, cannabis, and alcohol [15]. Some evidence cautiously suggests that emerging adults might be more likely to stay engaged in treatment when clinicians acknowledge their preferences and values [16,17]. However, there are concerns that individuals, including emerging adults, who are experiencing more severe mental health symptoms or have low health-related quality of life (HRQL) may have a diminished capacity to actively participate in complex mental health treatment decisions without support from a proxy decision maker or caregiver [18,19]. For instance, an online survey suggests that adults with anxiety disorders may require more psychosocial support to facilitate their decision-making needs [18]. The authors found that only 55% of adults 18–77 years old with anxiety disorders preferred using SDM during clinical encounters, while the remainder preferred a more passive role. Whether these results are generalizable to emerging adults remains unclear.

To our knowledge, no research has examined whether symptoms of anxiety and depression influence the level of engagement in SDM among emerging adults. Gaining a better understanding of the barriers and facilitators of SDM in emerging adults with these conditions may help guide future research, service development, and clinical practice.

### Study objectives and hypotheses

This study aimed to explore whether prevalent anxiety, depression symptoms or health-related quality of life (HRQL) are associated with the level of perceived SDM involvement during a clinical encounter in primary care or addiction and

mental health (AMH) settings in Alberta, Canada. We hypothesized that prevalent symptoms of anxiety or depression would be negatively associated with SDM (i.e., increased symptoms would yield decreased perceived SDM involvement). A secondary objective was to explore whether health-related quality of life (HRQL) would be positively associated with SDM (i.e., increased HRQL would yield increased SDM ratings). In a post-hoc analysis, we explored whether patient age was associated with SDM engagement. Furthermore, we hypothesized that age would be negatively associated with SDM.

## Materials and methods

### Overview of study design and setting

Between January and December 2019, a cross-sectional survey was conducted in person at six community-based primary care clinics and eight outpatient AMH clinics in urban, rural, and remote communities in Alberta, Canada. A cross-sectional study was an appropriate methodology to identify preliminary correlations, as research on SDM in emerging adults with anxiety and depression is in its infancy. This study design will be useful for generating hypotheses for exploration in future research. Alberta is a geographically large province with a single-payer health system that delivers health and mental health services to approximately 4.37 million residents as of 2019. A subset of data on emerging adults (18–29 years) and their corresponding HCPs was obtained from an overarching province-wide study on SDM in adults (18–64 years). The Health Research Ethics Board at the University of Alberta (application number: Pro00066937) provided ethical oversight and approvals for the study, and Alberta Health Services provided operational approval. The Strengthening of the Reporting of Observational Studies (STROBE)-Statement was used to guide the reporting of the results [20].

### Participants

Study participants included healthcare providers (HCPs) and their patients who met predetermined eligibility criteria. All participants took part voluntarily, and neither cohort received reimbursement or compensation for their involvement. Since the study was carried out at publicly funded health centres, the provincial healthcare system covered the costs of clinic visits for all patient participants.

**Eligibility criteria.** *Healthcare providers (HCPs)*

HCPs must have been licensed to practice in Alberta and actively employed at an Alberta Health Services community AMH or primary care setting to be eligible to participate. HCPs were required to provide informed, written consent, and understand, write, and speak English. There were no other exclusion criteria for HCPs.

*Patients*

Patients must have been 18–29 years old at the time of consent and voluntarily seeking health services at a participating primary care or outpatient mental health setting. They must be able to (a) provide informed, written consent, (b) participate without a proxy, and (c) understand, write, and speak English sufficiently to be eligible for participation. All patients must have active provincial health insurance and a personal health care number (PHN). Patients were eligible at any point during their treatment, whether at intake/first visit or follow-up visits.

Patients were ineligible if they were (a) currently experiencing a mental health crisis (defined as a clinically significant risk of harm to oneself or others within the last seven calendar days in the opinion of their HCP), or (b) whose clinic visit was mandated or coerced for any reason (e.g., drug-related pre-trial diversion program).

### Data collection

Purposive sampling was used to identify eligible study participants. Since this study used a subset of data from another study, a sample size calculation was not performed, and all eligible data were used.

**Recruitment process.** First, the study team met with staff and site managers at each clinic to obtain written permission to recruit interested HCPs and collect data at the study site. After obtaining informed consent, the HCPs completed a paper-based sociodemographic questionnaire. Participating HCPs then recruited and screened potentially eligible patients and referred those interested to the research assistant. Following the clinic visit with each participating patient, HCPs completed all remaining study questionnaires on paper. Each participating patient was required to complete a series of questionnaires on an encrypted electronic tablet immediately after the clinic visit. A sociodemographic form was used to gather information on participants' age, gender, marital status, ethnicity, education level, total household income, and any diagnosed chronic health conditions. Afterwards, patients completed questionnaires assessing SDM, HRQL, and symptoms of anxiety and depression. The research assistant provided support to participants as needed when using the tablet. All collected data were stored in the Health Research Data Repository at the University of Alberta's Faculty of Nursing. The data were anonymized and accessible only to the research team.

## Variables

**Shared decision-making.** The Alberta Shared Decision-MaKing Instrument (ASK-MI) measured healthcare providers' and patients' perceptions of SDM [21,22]. The ASK-MI is a 6-item dyadic questionnaire using a Likert scale (strongly agree to strongly disagree), with scores ranging from 6 to 36. The internal consistency (Cronbach's alpha) for the ASK-MI-patient version is 0.996, and the ASK-MI-healthcare provider version is 0.950 [21,22]. Both versions are completed independently by the patient and their HCP following the consultation. Neither the HCP nor the patient were aware of the results of the corresponding questionnaire. The ASK-MI scores of dyads were calculated by averaging the scores from the ASK-MI patient version and the ASK-MI HCP version. Values between 6 and 12 indicate "high" levels of SDM, between 13 and 24 indicate "moderate" levels of SDM, and values 25 and 36 indicate "low" levels of SDM. The questionnaires are provided in supplementary **S1** and **S2 Files**.

**Symptoms of anxiety and depression.** Symptoms of anxiety and depression were evaluated using the Hospital Anxiety and Depression Scale (HADS) [23,24]. The HADS is divided into two sections (HADS-A and HADS-D). Clinically significant anxiety symptoms were identified as scores of ≥8 on HADS-A, and clinically significant depression symptoms were identified as scores of ≥8 on HADS-D. Results from a systematic review indicate that these cut-off points have a high positive predictive value for detecting individuals with anxiety and depressive disorders, respectively [23]. Clinically significant symptoms were reported descriptively, and anxiety and depression symptoms were analyzed separately and as continuous variables.

**Health-related quality of life.** Health-related quality of life (HRQL) was assessed using the EuroQol-5D-5L (EQ-5D-5L) ("EuroQol Research Foundation. EQ-5D-5L User Guide. Available from: https://euroqol.org/publications/user-guides," 2019) and visual analogue scale (VAS) [25,26]. The VAS is measured on a continuous scale from 0 to 100, with 0 representing the worst health and 100 representing the best health.

**Data analysis.** Data were cleaned using Microsoft Excel (Microsoft Corporation, 2023) and analyzed with SPSS Version 23.0 (IBM Corp. Released 2015. IBM SPSS Statistics for Windows, Version 23.0. Armonk, NY: IBM Corp.). Pearson correlations were performed to explore relationships between key variables, with the significance level set at $P = 0.05$. Due to the small sample size, any missing data were imputed using the mean score. Mean (SD) differences between HCPs and patients were reported. T-tests were used to determine if patients and HCPs reported different perceptions of SDM during the clinic visit. A post hoc analysis was conducted to investigate whether a relationship existed between patient age and SDM ratings.

## Results

In the overarching study, 34 potentially eligible healthcare clinics were invited to participate, and 20 (58.8%) agreed. Of these 20 sites, 14 (70.0%) collected data on emerging adults and were thus eligible for the present study. Six of 11 (54.5%) sites were primary care settings, and eight of nine sites (88.9%) were community AMH settings (Fig 2).

**Study flow chart**

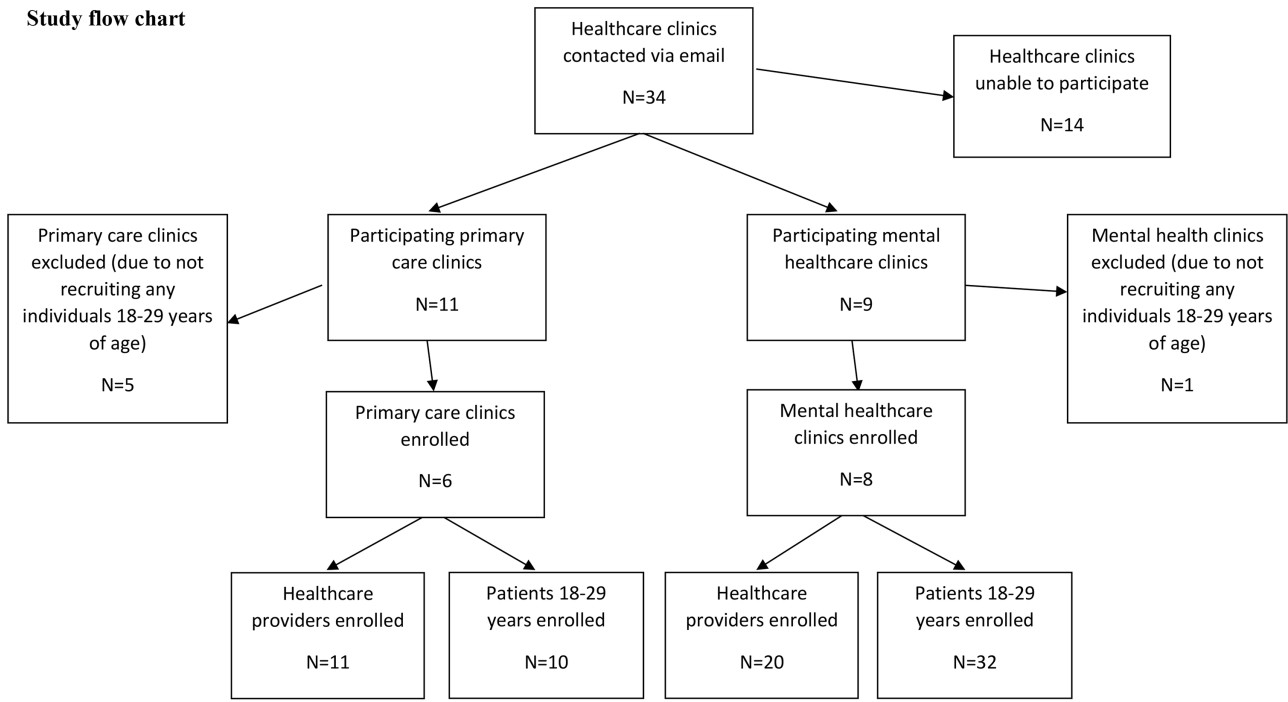

**Fig 2. Study flowchart of participant inclusion and exclusion.** Study flow diagram showing the number of primary care and community addiction and mental health clinics contacted, enrolled, and recruiting 18–29-year-old patients, as well as the number of patients and healthcare providers who ultimately participated in the study.

## Participant characteristics

**Healthcare providers.** Thirty-one healthcare providers (HCPs) with ages ranging from 27 to 58 years were recruited. Twenty HCPs (64.5%) came from mental health clinics, while 11 (34.4%) were from primary care settings. Most HCPs were women (n = 24, 77.4%), including mental health therapists (e.g., counsellors or psychologists) (n = 8, 25.8%) and primary care physicians (n = 7, 22.6%). Table 1 displays the sociodemographic characteristics of HCP participants.

**Patients.** Forty-two patients, 18−29 years, were recruited; 10 (23.8%) were sampled from primary care settings, and 32 (76.2%) were sampled from community addiction and mental health settings. The mean age of the patients was 24.1 years (SD, +/- 3.0 years). HADs criteria for symptoms of anxiety (HADS-A ≥ 8) were met for 73.8% (n = 31) of the patients, followed by 40.5% (n = 17) were positive for symptoms of depression (HADS-D ≥ 8), 38.1% (n = 16) met the criteria for both anxiety and depression, and 26.2% (n = 11) were negative for either condition. The mean patient ASK-MI score was 8.07 (SD, +/- 2.83), the mean HCP ASK-MI score was 9.31 (SD, +/- 3.30), and the mean dyad score was 8.69 (SD, +/- 2.01). An independent samples t-test revealed a marginally significant difference in ASK-MI scores between patients and HCPs (mean difference = −1.24, df = 82, p = 0.069, two-tailed). Table 2 displays the sociodemographic characteristics of the patient participants.

## Main findings

After performing Pearson R correlations, symptoms of anxiety, depression and HRQL were not significantly correlated with the level of SDM engagement during the clinic visit. However, post hoc analyses revealed that patient age was negatively correlated with perceived SDM among the dyad ratings; R = −0.34, p = 0.03. A sensitivity analysis showed symptoms of anxiety were negatively correlated with HRQL, R = 0.50, p < 0.001, and symptoms of depression were negatively correlated with HRQL, R = 0.61, p < 0.001. Table 3 shows the correlation matrix of the main study variables.

**Table 1. Healthcare provider sociodemographic characteristics.**

| Variable | Value |
|---|---|
| ***Demographic characteristics*** | |
| N participants | 31 |
| Age, years (mean ± SD) | 40.9 (8.58) |
| Age, range (years) | 27-58 |
| Gender identity, n (%) man | 7 (22.6) |
| **Clinical setting n (%)** | |
| Primary care | 11 (35.4) |
| Addiction and mental health | 20 (64.5) |
| **Setting location by Alberta Health Zone n (%)** | |
| North | 5 (16.1) |
| Edmonton | 14 (45.2) |
| Central | 3 (9.7) |
| Calgary | 7 (22.6) |
| South | 2 (6.5) |
| **Profession n (%)** | |
| Physicians | 7 (22.6) |
| Registered nurses | 4 (12.9) |
| Mental health therapists (e.g., counsellor or psychologist) | 8 (25.8) |
| Social workers | 5 (16.1) |
| Dieticians | 3 (9.68) |
| Physical or occupational therapists | 4 (12.9) |

## Discussion

To our knowledge, this is the first study to explore whether patient psychosocial characteristics, such as HRQL and symptoms of anxiety or depression among emerging adults, are associated with the perceived level of SDM engagement during a clinic visit at a mental health or primary care setting. Contrary to our hypotheses, we did not observe any association between symptoms of anxiety, depression or HRQL with SDM in this study. To better understand these findings, we conducted post hoc analyses exploring whether additional patient characteristics, such as age, may be associated with the dyadic level of SDM. We found that patient age showed a weak but statistically significant negative (i.e., inverse) association with SDM. Considering only a narrow age range of individuals, emerging adults (18–29 years) were included in the analysis; the interpretation of this finding is unclear and warrants further investigation.

Considering most perceived SDM scores were within the "excellent" (i.e., upper) range, these results may suggest the participants were able to fully participate in SDM with their HCPs regardless of their symptoms of anxiety, depression, or HRQL. This study provides preliminary evidence that mild to moderate symptoms of anxiety, depression, or HRQL are not significant barriers to engaging in SDM among emerging adults in outpatient mental health or primary care settings. Our main findings are consistent with previous research suggesting individuals (including emerging adults) with mental health issues may be able to participate in mental health-related treatment decisions [10,12], aligning with an SDM approach. Previous research also suggests that adults with depression in primary care can engage in SDM; however, trust (i.e., therapeutic alliance) between the patients and HCPs may be a key factor in facilitating SDM [27,28].

**Table 2. Patient characteristics.**

| Variable | |
|---|---|
| *Demographic characteristics* | Value |
| N participants | 42 |
| Age, years (mean ± SD) | 24.1 (3.0) |
| Gender identity, n (%) man | 9 (22.0) |
| Missing, n (%) | 2 (4.89) |
| Ethnicity, Caucasian n (%) | 23 (54.8) |
| 1 ≥ reported chronic health conditions n (%) | 32 (76.2) |
| Missing, n (%) | 1 (2.4) |
| **Marital status n (%)** | |
| Legally married or common law | 7 (16.7) |
| Missing, n (%) | 2 (4.8) |
| **Total family income quintile n (%)** | |
| Less than $20,000 | 11 (26.2) |
| $20,000 to $34,999 | 4 (9.5) |
| $35,000 to $49,999 | 4 (9.5) |
| $50,000 to $74,999 | 6 (14.3) |
| $75,000 to $99,999 | 6 (14.3) |
| $100,000 to $149,999 | 3 (7.1) |
| $150,000 or more | 1 (2.4) |
| Missing | 7 (16.7) |
| **Employment** | |
| Employed, working 40 or more hours per week | 9 (21.4) |
| Employed, working 1–39 hours per week | 14 (33.3) |
| Not employed, looking for work | 8 (19.0) |
| Not employed, not looking for work | 2 (4.8) |
| Retired | 0 |
| Unable to work | 9 (21.4) |
| **Education** | |
| Less than a high school degree | 5 (11.9) |
| High school degree or equivalent | 13 (31.0) |
| Some post-secondary education but no degree | 13 (31.0) |
| Registered apprenticeship or other trades certificates or diploma | 3 (7.1) |
| Associate degree | 1 (2.4) |
| Bachelor degree | 6 (14.3) |
| Graduate degree | 1 (2.4) |
| **Clinical setting** | |
| Primary care | 10 (23.8) |
| Addiction and mental health | 32 (76.2) |
| **Setting location by Alberta Health Zone** | |
| North | 7 (16.7) |
| Edmonton | 21 (50.0) |
| Central | 3 (7.1) |
| Calgary | 9 (21.4) |
| South | 2 (4.8) |

*(Continued)*

**Table 2.** (Continued)

| Variable | |
|---|---|
| **Clinical outcomes** | |
| HADS ≥8 anxiety n (%) | 31 (73.8) |
| HADS ≥8 depression n (%) | 17 (40.5) |
| HADS ≥8 anxiety and depression n (%) | 16 (38.1) |
| EQ-5D dimension index (mean ± SD) | 0.72 (0.21) |
| EQ-5D VAS (mean ± SD) | 68.3 (18.1) |
| PHE median score ≥ 3 n (%) | 25 (59.5) |

**Table 3. Correlation matrix of study variables.**

| | | Patient age | Patient education attainment | Patient anxiety (HADS-A) | Patient depression (HADS-D)) | Patient HRQL (EQ5D-VAS) | SDM (ASK-MI, mean dyad) |
|---|---|---|---|---|---|---|---|
| **Patient age** | *Pearson R correlation* | 1 | .574** | −.185 | −.148 | .056 | −.335* |
| | *Sig. (2-tailed)* | | <.001 | .242 | .350 | .724 | .030 |
| | *N* | 42 | 42 | 42 | 42 | 42 | 42 |
| **Patient education attainment** | *Pearson R correlation* | .574** | 1 | −.227 | −.279 | .126 | −.310* |
| | *Sig. (2-tailed)* | <.001 | | .148 | .074 | .427 | .046 |
| | *N* | 42 | 42 | 42 | 42 | 42 | 42 |
| **Patient anxiety (HADS-A)** | *Pearson R correlation* | −.185 | −.227 | 1 | .648** | −.504** | −.033 |
| | *Sig. (2-tailed)* | .242 | .148 | | <.001 | <.001 | .833 |
| | *N* | 42 | 42 | 42 | 42 | 42 | 42 |
| **Patient depression (HADS-D)** | *Pearson R correlation* | −.148 | −.279 | .648** | 1 | −.605** | .027 |
| | *Sig. (2-tailed)* | .350 | .074 | <.001 | | <.001 | .866 |
| | *N* | 42 | 42 | 42 | 42 | 42 | 42 |
| **Patient HRQL (EQ5D-VAS)** | *Pearson R correlation* | .056 | .126 | −.504** | −.605** | 1 | −.222 |
| | *Sig. (2-tailed)* | .724 | .427 | <0.001 | <0.001 | | .158 |
| | *N* | 42 | 42 | 42 | 42 | 42 | 42 |
| **SDM (ASK-MI, mean dyad)** | *Pearson R correlation* | −.335* | −310* | −.033 | .027 | −.222 | 1 |
| | *Sig. (2-tailed)* | .030 | .046 | .833 | .866 | .158 | |
| | *N* | 42 | 42 | 42 | 42 | 42 | 42 |

## Limitations

Several limitations in this study warrant a cautious interpretation of the results. First, a cross-sectional design cannot ascertain a temporal or causative relationship between the study variables; therefore, only hypotheses can be generated from the data in this study. However, longitudinal data were not necessary to answer our research questions, which were to explore whether any association exists at baseline between SDM and HRQL, anxiety, or depression. The lack of longitudinal data, however, precludes any ability to observe the potential associations between study variables and SDM over time. Future longitudinal studies will be needed to assess the barriers and facilitators of SDM in emerging adults with mental health concerns.

Additionally, it will be essential to consider prior patient relationships with the HCPs. HCPs who have known their patient for longer may have developed a more therapeutic alliance and relationship with that patient, which may lead to a higher SDM score. Future research could consider enrolling new patients and providers, as compared to those with a more

 

longstanding relationship, while ensuring visit length remains the same. Next, we only collected a relatively modest sample size as only a few individuals aged 18–29 participated in the larger parent study on SDM [3,16]. The small sample size precluded a comparison of outcomes between primary care and community AMH settings, which may be of interest in future research. For example, the duration of the clinical interaction may vary between settings, potentially leading to a change in the perception of the extent to which SDM occurred during the clinic visit. Limited demographic data prevented the exploration of additional variables, such as gender identity, race/ethnicity, and socioeconomic status, on perceived levels of SDM.

Additionally, participants (patients and HCPs) who were more likely to value an SDM approach may have been more willing to participate, leading to selection and performance biases. A ceiling effect was observed in the SDM dyad and SDM HCP ratings, possibly due to performance bias from the study participants. For instance, HCP participants knew in advance that the researchers would be evaluating SDM. Therefore, clinicians may have been careful to perform SDM according to our criteria, leading to high scores, and patients may have felt a need to give their HCP "high marks". However, the marginal difference found between HCP and patient scores may suggest that HCPs were inclined to report higher SDM scores, while patients may have been slightly more critical in their assessment.

Moreover, the ASK-MI instruments had very high Cronbach Alpha scores, which may indicate some redundancy in the questionnaire items. More research is needed to explore why HCPs perceived SDM to be higher than patients, and whether the limitations observed in this study would be better addressed by tackling performance bias and selection bias, or if the instrument should be redesigned to reduce the ceiling effect. For example, patients and HCPs could evaluate each other's level of SDM involvement, rather than assessing SDM more generally, which may reduce the likelihood of HCPs rating the SDM that occurred as "excellent" or high. Future iterations of the instrument should consider removing the language "excellent" or "poor" from the tool, as this may inadvertently signal to the HCP that they do not perform well in the session with their client. Still, there are justifiable reasons why SDM ratings might be low, even when the quality of care remains high or satisfactory (e.g., lack of time to perform SDM, a patient's mental health crisis, or the patient prefers the healthcare provider to make decisions) [29].

## Conclusions

SDM is a complex and often iterative process characterized by the balance of patient and HCP involvement in decision-making, informed by the best available evidence. This study suggests emerging adults may be able to engage in high levels of SDM with their HCP regardless of anxiety/depression or HRQL. However, several sources of bias in this study reduce confidence in the observed effect. We understand SDM to exist on a continuum, and many factors may facilitate or impede the level of perceived SDM during a clinic visit among emerging adults and their HCPs, which could not be accounted for in this study. The findings of this study should inform future research on SDM.

## Supporting information

**S1 File. ASK-MI HCP version.**
(PDF)

**S2 File. ASK-MI PAT version.**
(PDF)

**S3 File. Anonymized study data.**
(XLSX)

## Acknowledgments

The authors thank Kevin Challacombe, Janet Lam, and Ariana Lewis for their assistance with data and reference management. We also appreciate the participants and the site operators for their help with recruitment and data collection.

## Author contributions

**Conceptualization:** Tyler Marshall, Karin Olson, Adam Abba-Aji, Xin-Min Li, Richard Lewanczuk, Sunita Vohra.

**Data curation:** Tyler Marshall.

**Formal analysis:** Tyler Marshall, Karin Olson.

**Funding acquisition:** Tyler Marshall, Karin Olson, Sunita Vohra.

**Investigation:** Tyler Marshall, Karin Olson, Richard Lewanczuk, Sunita Vohra.

**Methodology:** Tyler Marshall, Karin Olson, Adam Abba-Aji, Xin-Min Li, Richard Lewanczuk, Sunita Vohra.

**Project administration:** Tyler Marshall.

**Supervision:** Karin Olson, Sunita Vohra.

**Validation:** Karin Olson, Adam Abba-Aji, Xin-Min Li, Richard Lewanczuk, Sunita Vohra.

**Visualization:** Tyler Marshall.

**Writing – original draft:** Tyler Marshall.

**Writing – review & editing:** Tyler Marshall, Karin Olson, Adam Abba-Aji, Xin-Min Li, Richard Lewanczuk, Sunita Vohra.

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
