## [Decision Letter · Decision Letter 0]

13 Nov 2023

Dear Dr. Vohra,

Thank you for submitting your manuscript to PLOS ONE. After careful consideration, we feel that it has merit but does not fully meet PLOS ONE’s publication criteria as it currently stands. Therefore, we invite you to submit a revised version of the manuscript that addresses the points raised during the review process.

We look forward to receiving your revised manuscript.

Kind regards,

Yaara Zisman-Ilani

Academic Editor

PLOS ONE

Journal Requirements:

4. We notice that your supplementary tables are included in the manuscript file. Please remove them and upload them with the file type 'Supporting Information'. Please ensure that each Supporting Information file has a legend listed in the manuscript after the references list.

**Additional Editor Comments:**

Dear Drs. Marshall and Vohra,

Thank you for your submission to PLOS ONE. Your manuscript was sent to two expert reviewers, both provided thorough feedback and asked for major revisions. Their detailed feedback is attached. Given my familiarity with the field of SDM in mental health, I would like to offer additional feedback specifically related to your Introduction and Discussion sections. Please ensure your arguments align with current and updated work in SDM in mental health. For example, when you write that SDM is associated with better satisfaction and adherence, you referenced SDM studies not in the mental health field, although there are several reviews about SDM intervention in mental health. Another example is in the Discussion section; the opening paragraph (and subsequent discussion) appears to inaccurately present SDM in mental health in the context of medication decisions only, neglecting the broader spectrum of SDM in mental health interventions, such as those in recovery, psychiatric rehabilitation, self determination, and social prescribing (e.g., employment, housing). 

Reviewers' comments:

Reviewer's Responses to Questions

**Comments to the Author**

1. Is the manuscript technically sound, and do the data support the conclusions?

Reviewer #1: Partly

Reviewer #2: Partly

2. Has the statistical analysis been performed appropriately and rigorously?

Reviewer #1: No

Reviewer #2: Yes

3. Have the authors made all data underlying the findings in their manuscript fully available?

Reviewer #1: Yes

Reviewer #2: No

4. Is the manuscript presented in an intelligible fashion and written in standard English?

Reviewer #1: Yes

Reviewer #2: Yes

Reviewer #1: This manuscript explores the relationship between shared decision-making, patient health engagement, health-related quality of life, and anxiety and depression symptoms in emerging young adults. This topic is very important and understudied. However, the manuscript is limited by some analytical decisions and a lack of clarity around the research question and hypothesis. It is my recommendation that the paper could be substantially improved by re-analyzing the data as continuous rather than categorical and re-writing the introductory framing and conclusion of the paper. My specific suggestions are below:

1. The authors provide a strong and clear discussion of the four measures used in this paper: ASK-MI, PHE-S, HADS, and HRQL. However, each of these measures is a continuous variable that they chose to dichotomize/make categorical. The authors do not describe the rationale for dichotomizing the data, and dichotomizing the data limits the potential variance that could be measured if the measures were included as continuous variables. The authors highlight in their limitations that there is limited variance, particularly in the ASK-MI, as none of the scores fell into the “unacceptable” range. Additionally, the decision to dichotomize the HRQL data based on the average national data could be questioned. The limited variance and dichotomized variables makes the results challenging to interpret. The paper would be substantially strengthened by using each of these variables in their initial continuous form.

2. The research questions and hypothesis proposed in the current study are not clear as a reader. In the “objectives” section of the abstract they note that they are exploring a) SDM association with PHE and HRQL and b) PHE associated with HRQL. They do not mention anxiety/depression in the objective or in the title of the paper. However, in the results section, they describe a main result of symptoms of anxiety/depression negatively related to PHE that was not part of a priori objective. Similarly, no hypothesis were provided as to the direction of the effect, and it is not clear what theory the authors have about these relationships a priori. Ideally, all null results will also be presented in the abstract.

3. Additionally, it may improve the clarity of the paper to include a theoretical model of how the four concepts (SDM, PHE, HRQL, A/D) relate to one another. It struck me that the initial objective proposed may describe a mediator model, and that the paper may also be strengthened by considering whether the authors are hoping to conduct a mediator model (SDM PHE HRQL). However, this does not include anxiety/depression, which appears to be a major interest for the authors.

4. It would similarly be helpful to clarify what theoretical gap this paper is filling and how the sample relates to that. The authors write: “To our knowledge, no investigations have explored whether an association exists between SDM, PHE, and HRQL among emerging adults with anxiety and depression” (page 4), but my understanding is that their participants include a general sample of individuals receiving physical/mental health services with and without anxiety/depression. As such, it should be clarified that this study is not only including individuals with A/D. Additionally, it may help clarity of the paper to provide additional information about what conditions the participants are presenting for and to note clearly that, per consort chart, a majority (approx. 76%) of participants are in mental health versus primary care settings. The title may imply even numbers of participants in these settings.

5. It is recommended that the conclusion of the paper be rewritten to more clearly outline the results and their implications. Importantly, the current interpretation of results in the discussion includes a number of statements that may not be supported by the evidence. First, they state the negative relationship between anxiety/depression and PHE implies that promotion of PHE could help reduce symptoms of depression and anxiety. This is a correlational relationship, and it seems just as likely that reducing symptoms of depression and anxiety may increase PHE. Second, the authors note that their findings are consistent with other studies suggesting a relationship between anxiety/depression and PHE, but also state they are the first study on this topic. It should be clarified how the current study fits into the existing literature. There are also a couple of times throughout the conclusion when related literature is described but it is not clear how this relates to the current study. For example, the authors suggest that cognitive impairment may be one reason for relationship between PHE and anxiety/depression and it is unclear why only one variable was highlighted, since cognitive impairment was not studied or discussed in the background literature. Similarly, the 4th paragraph of the conclusion begins with a topic sentence discussing PHE/SDM but the rest of the paragraph does not refer to SDM.

6. The study flow chart and tables are particularly helpful for understanding the study participants and analysis/results. It would be helpful to relabel table so include more specific labels than the “value” at the top. Additionally, percent signs may be helpful and df may be included for the Chi squared analysis.

Some more minor points are as follows:

- In the background of the abstract (page 3), the authors state people with anxiety and depressive disorders may respond better to SDM approaches. It should be clarified whether the authors are saying that they respond better to SDM approaches as compared to other approaches or as compared to other populations.

- The description of 31 healthcare providers and 42 emerging adult patient dyads may be more clearly described as: “42 dyads (31 unique healthcare providers)”

- The keywords for the article do not seem to reflect the major point of the paper. For example, it is recommended that shared decision-making and patient engagement (versus patient centered care) are included as keywords. Anxiety and depression should be included only if this is key to the authors research questions (see comments above).

- On page 4, the authors cite a recent systematic review that suggests that SDM may benefit adults with anxiety and depression. Please clarify in what way SDM benefits.

- On page 4, the authors refer to “younger individuals” with mental illnesses. Please pick one term, such as “emerging adults” to use throughout the introduction for clarity.

- The limitations section should only include limitations of the study and the first paragraph indicating “strengths” of the study can be cut

Reviewer #2: Shared decision-making is gaining prominence in the mental health field, and I found the manuscript to be quite captivating. Below, I have outlined my comments regarding the key points I observed. I hope that this will be valuable for enhancing the paper.

1. The introduction made it clear that the primary focus was on examining the relationship between Shared Decision-Making (SDM) and factors such as anxiety or depressive symptoms, patient health engagement, and health-related quality of life. However, the hypotheses formulated by the authors in this study were not clearly stated. Please articulate the hypotheses in this study clearly and provide a coherent explanation for the necessity of these hypotheses."

2. It was mentioned that 31 healthcare professionals and 42 patients participated. How many individuals declined to participate� How many individuals did not meet the inclusion criteria and were excluded? I propose providing a detailed description of how many individuals were approached for both healthcare professionals and patients, and as a result, 31 healthcare professionals and 42 patients were included.

3. I understood that pages 8, lines 1 to 6, contain descriptions regarding the responses of dyads, referring to pairs of patients and healthcare professionals. However, in Table 3, apart from these dyad results, the top two rows display results for 31 healthcare professionals and 42 patients.

I am curious about the relationship between the top two rows in Table 3 (results for 31 healthcare professionals and 42 patients) and the third row (results for Dyads of patients and healthcare professionals). It would be helpful for better understanding if you could explain the details of what is presented in Table 3 in the text, row by row.

4. On page 8, in the 9th line, it is stated 'Table 4 shows the association with PHE and HRQL among emerging adults.' However, upon reviewing Table 4, it is evident that the Table 4 reports the association between SDM and anxiety and/or depression, HRQL, and treatment setting, rather than the association between SDM, PHE, and HRQL. Please correct this discrepancy to ensure alignment between the information presented in the Table 4 and the explanation provided in the text."

5. Related to this, on page 8, in the 9th line, it states, 'No significant relationship was observed between SDM and PHE, r=0.03, P=0.83.' However, this information cannot be inferred from Table 4.

6. There were criteria provided in the main text for categorizing HRQL into 'Normal' and 'Abnormal.' However, for anxiety and depression, there is no explanation in the main text regarding the criteria for categorizing into 'Normal' and 'Abnormal,' yet these classifications are used in Table 4. Please provide an explanation for the criteria in the main text."

7. On page 8, in the 15th line, it is stated 'Table 5 shows the association between PHE and HRQL.' However, upon reviewing Table 5, it is evident that the table not only displays the association between PHE and HRQL but also the association between PHE, SDM, and anxiety and/or depression. I propose correcting the description to accurately reflect the contents of the table.

8. In the results, please clearly describe whether the formulated hypotheses were supported or rejected based on the study results. Based on that, in the Discussion section, please discuss the factors that contributed to the support or rejection of the hypotheses.

**Do you want your identity to be public for this peer review?** For information about this choice, including consent withdrawal, please see our Privacy Policy

Reviewer #1: No

Reviewer #2: No

---

## [Author Response · Author response to Decision Letter 1]

27 Jul 2024

We have double-checked the requirements and have used the style template as requested.

2. We note that the grant information you provided in the ‘Funding Information’ and ‘Financial

Disclosure’ sections do not match. When you resubmit, please ensure that you provide the correct grant numbers for the awards you received for your study in the ‘Funding Information’ section.

Thank you, this has been corrected in the system to state the following:

“This research has been funded by the generous support of the Stollery Children’s Hospital Foundation through the Women and Children’s Health Research Institute. This research was also funded by the Faculty of Graduate Studies and Research (RES0033014) at the University of Alberta.”

Upon re-submitting your revised manuscript, please upload your study’s minimal underlying data set as either Supporting Information files or to a stable, public repository and include the relevant URLs, DOIs, or accession numbers within your revised cover letter. For a list of acceptable repositories, please see

http://journals.plos.org/plosone/s/data-availability#loc-recommended-repositories. Any potentially identifying patient information must be fully anonymized.

Important: If there are ethical or legal restrictions to sharing your data publicly, please explain these restrictions in detail. Please see our guidelines for more information on what we consider unacceptable restrictions to publicly sharing data: http://journals.plos.org/plosone/s/data-availability#loc-

unacceptable-data-access-restrictions. Note that it is not acceptable for the authors to be the sole named individuals responsible for ensuring data access. We will update your Data Availability statement to reflect the information you provide in your cover letter.

We have confirmed with our ethics board that we are able to provide a minimal, deidentified dataset of the primary outcome measures which can be found in the supporting information.

4. We notice that your supplementary tables are included in the manuscript file. Please remove them and upload them with the file type 'Supporting Information'. Please ensure that each Supporting Information file has a legend listed in the manuscript after the references list.

We have removed the supplementary tables from the manuscript and have included them as supplementary information as instructed.

Additional Editor Comments:

Dear Drs. Marshall and Vohra,

Thank you for your submission to PLOS ONE. Your manuscript was sent to two expert reviewers, both provided thorough feedback and asked for major revisions. Their detailed feedback is attached.

Given my familiarity with the field of SDM in mental health, I would like to offer additional feedback specifically related to your Introduction and Discussion sections. Please ensure your arguments align with current and updated work in SDM in mental health. For example, when you write that SDM is associated with better satisfaction and adherence, you referenced SDM studies not in the mental health field, although there are several reviews about SDM intervention in mental health. Another example is in the Discussion section; the opening paragraph (and subsequent discussion) appears to inaccurately present SDM in mental health in the context of medication decisions only, neglecting the broader spectrum of SDM in mental health interventions, such as those in recovery, psychiatric rehabilitation, self determination, and social prescribing (e.g., employment, housing).

Thank you, we agree and have performed a major revision to the manuscript (almost entirely rewritten in many places) to be more up-to-date with current literature, inclusive of the broader spectrum of SDM in a variety of mental health interventions.

We have cited and discussed the following additional citations in the introduction and discussion.

Schauer, C., Everett, A., Del Vecchio, P., & Anderson, L. (2007). Promoting the value and practice of shared decision-making in mental health care. Psychiatric Rehabilitation Journal, 31(1), 54.

Drake, R. E., Cimpean, D., & Torrey, W. C. (2009). Shared decision making in mental health: prospects for personalized medicine. Dialogues in clinical neuroscience, 11(4), 455-463.

Reviewers' comments:

Reviewer's Responses to Questions

Comments to the Author

1. Is the manuscript technically sound, and do the data support the conclusions?

Reviewer #1: Partly

Reviewer #2: Partly

2. Has the statistical analysis been performed appropriately and rigorously?

Reviewer #1: No

Reviewer #2: Yes

3. Have the authors made all data underlying the findings in their manuscript fully available?

Reviewer #1: Yes

Reviewer #2: No

4. Is the manuscript presented in an intelligible fashion and written in standard English?

Reviewer #1: Yes

Reviewer #2: Yes

5. Review Comments to the Author

Reviewer #1:

This manuscript explores the relationship between shared decision-making, patient health engagement, health-related quality of life, and anxiety and depression symptoms in emerging young adults. This topic is very important and understudied. However, the manuscript is limited by some analytical decisions and a lack of clarity around the research question and hypothesis. It is my recommendation that the paper could be substantially improved by re-analyzing the data as continuous rather than categorical and re-writing the introductory framing and conclusion of the paper.

My specific suggestions are below:

1. The authors provide a strong and clear discussion of the four measures used in this paper: ASK-MI, PHE-S, HADS, and HRQL. However, each of these measures is a continuous variable that they chose to dichotomize/make categorical. The authors do not describe the rationale for dichotomizing the data, and dichotomizing the data limits the potential variance that could be measured if the measures were included as continuous variables. The authors highlight in their limitations that there is limited variance, particularly in the ASK-MI, as none of the scores fell into the “unacceptable” range. Additionally, the decision to dichotomize the HRQL data based on the average national data could be questioned. The limited variance and dichotomized variables makes the results challenging to interpret. The paper would be substantially strengthened by using each of these variables in their initial continuous form.

We had dichotomized the data based on clinical significance to try and promote the clinical utility of the results (i.e., so the results may be more clinically meaningful). However, we agree that analyzing these variables on a continuous scale may provide some additional statistical power, and we have reanalyzed the data as suggested. To calculate dyadic SDM (ASK-MI) scores, we used the mean score between the healthcare provider and the patient.

2. The research questions and hypothesis proposed in the current study are not clear as a reader. In the “objectives” section of the abstract they note that they are exploring a) SDM association with PHE and HRQL and b) PHE associated with HRQL. They do not mention anxiety/depression in the objective or in the title of the paper. However, in the results section, they describe a main result of symptoms of anxiety/depression negatively related to PHE that was not part of a priori objective. Similarly, no hypothesis were provided as to the direction of the effect, and it is not clear what theory the authors have about these relationships a priori. Ideally, all null results will also be presented in the abstract.

We agree and have clarified our research questions with an accompanying hypothetical model, including a directional effect. Please note that this change, along with changing how SDM is measured, has resulted in a major revision of the manuscript with new results.

Revised research objective:

“The primary objective of this study was to explore whether symptoms of anxiety, depression or HRQL were associated with the level of dyadic SDM reported after a clinical encounter among emerging adults who obtained treatment in a primary care or mental health setting in Alberta, Canada. We hypothesized that symptoms of anxiety or depression would be negatively associated with SDM (i.e., increased symptoms would yield decreased SDM ratings), and HRQL would be positively associated with SDM (i.e., increased HRQL would yield increased SDM ratings). Figure 1 displays the hypothetical model. In a post-hoc analysis, we explored whether patient age is associated with dyadic SDM. Here, we hypothesized age would be negatively associated with SDM.”

3. Additionally, it may improve the clarity of the paper to include a theoretical model of how the four concepts (SDM, PHE, HRQL, A/D) relate to one another. It struck me that the initial objective proposed may describe a mediator model, and that the paper may also be strengthened by considering whether the authors are hoping to conduct a mediator model (SDM PHE HRQL). However, this does not include anxiety/depression, which appears to be a major interest for the authors.

Thank you for this comment. In response to our research question clarification, we have included a hypothetical model. Since we collected cross-sectional data, we have evidence of anxiety/depression during and shortly prior to the clinic visit and SDM scores corresponding to the clinic visit. We are unable to explore the impact of SDM on anxiety, depression, or HRQL after the appointment, but rather the reverse: exploring whether anxiety/depression and HRQL impact the level of SDM engagement that occurs during the clinic visit.

4. It would similarly be helpful to clarify what theoretical gap this paper is filling and how the sample relates to that. The authors write: “To our knowledge, no investigations have explored whether an association exists between SDM, PHE, and HRQL among emerging adults with anxiety and depression” (page 4), but my understanding is that their participants include a general sample of individuals receiving physical/mental health services with and without anxiety/depression.

Thank you, as a result of the above clarifications of the research question and analysis, we were able to fill a theoretical gap: (i) who SDM may be most appropriate for and (ii) what symptoms are barriers and facilitators for achieving SDM in clinical encounters. Currently, most research explores the impact of the intervention (SDM) on health outcomes, but not as SDM as an outcome itself, which we suspect may vary depending on the healthcare provider, healthcare setting and patient’s conditions. It is important to understand in what settings and for what conditions SDM is most appropriate/effective, which is why we have now decided to explore the impact of anxiety/depression, and HRQL on the level of SDM.

As such, it should be clarified that this study is not only including individuals with A/D. Additionally, it may help clarity of the paper to provide additional information about what conditions the participants are presenting for and to note clearly that, per consort chart, a majority (approx. 76%) of participants are in mental health versus primary care settings. The title may imply even numbers of participants in these settings.

We have clarified in the text of the manuscript that the same is among emerging adults with and without anxiety and depression.

5. It is recommended that the conclusion of the paper be rewritten to more clearly outline the results and their implications. Importantly, the current interpretation of results in the discussion includes a number of statements that may not be supported by the evidence. First, they state the negative relationship between anxiety/depression and PHE implies that promotion of PHE could help reduce symptoms of depression and anxiety. This is a correlational relationship, and it seems just as likely that reducing symptoms of depression and anxiety may increase PHE.

Thank you, we removed PHE from the analysis this variable was not necessary for addressing our research question.

Second, the authors note that their findings are consistent with other studies suggesting a relationship between anxiety/depression and PHE, but also state they are the first study on this topic. It should be clarified how the current study fits into the existing literature. There are also a couple of times throughout the conclusion when related literature is described but it is not clear how this relates to the current study. For example, the authors suggest that cognitive impairment may be one reason for relationship between PHE and anxiety/depression and it is unclear why only one variable was highlighted, since cognitive impairment was not studied or discussed in the background literature. Similarly, the 4th paragraph of the conclusion begins with a topic sentence discussing PHE/SDM but the rest of the paragraph does not refer to SDM.

Thank you, we agree other studies have been conducted around shared decision-making in mental health but this study is novel as it explores predictors of shared decision-making in emerging adults with primary care or mental health settings. This has not been done before to our knowledge.

6. The study flow chart and tables are particularly helpful for understanding the study participants and analysis/results. It would be helpful to relabel table so include more specific labels than the “value” at the top. Additionally, percent signs may be helpful and df may be included for the Chi squared analysis.

Thank you, as the n (%) are located on the columns to the left and not every row reports the same statistic, that “value” is the most appropriate name for the right column on table 1. Since we have now reran the analysis using continuous values we are no longer using the chi-square

---

## [Editor Report · Decision Letter 1]

13 Nov 2024

PONE-D-23-23994R1

Predictors of shared decision-making among treatment-seeking emerging adults in primary care and community addiction and mental health settings: a cross-sectional study

PLOS ONE

Dear Authors,

Thank you for submitting your manuscript to PLOS ONE. After careful consideration, we have decided that your manuscript does not meet our criteria for publication and must therefore be rejected.

What significant novelty is added by this study? There is lack of clear conceptualization and evidence from scientific theory is also missing. Particularly,

How could a cross sectional study be explored? Instead use better terminology.What is the gap in research that you want to uncover? Which body of literature suggest that not a single study is conducted on predictors of shared decision making among emerging adulthood?What do you mean by “it is unclear whether prevalent symptoms of anxiety, depression and health-related quality of life (HRQL) are associated with the level of SDM that occurs during a clinical encounter”? Elaborate in what way?Since you were exploring something new, why you did not carry out a qualitative analysis initially? What a cross sectional quantitative survey was conducted?Lack of adequate literature is found in the introduction section to support your hypotheses.The internal consistency (Cronbach’s alpha) for the ASK-MI-patient version is 0.996, and the ASK-MI-healthcare provider version [6] is 0.950. This reliability is considered redundant and questionable.The sample size is too small for a cross sectional design and does not support the study generalizability.The study interpretation of results needs in-depth work both in terms of statistics and explanation

I am sorry that we cannot be more positive on this occasion, but hope that you appreciate the reasons for this decision.

Kind regards,

Sana Younas

Academic Editor

PLOS ONE

- - - - -

---

## [Author Response · Author response to Decision Letter 2]

18 Feb 2025

Thank you for the opportunity to reconsider our manuscript and for the helpful comments which we addressed below point-by-point. These responses can also be found in the original appeal letter.

1. Significance and novelty

Our study is the first to explore the relationship between shared decision-making (SDM), patient health engagement (PHE), and health-related quality of life (HRQL) among emerging adults (18–25 years) in outpatient primary care and mental health settings. While previous research has demonstrated the effectiveness of SDM in mental health contexts, our study contributes novel findings by focusing on an underexplored population, emerging adults, and by investigating associations with validated, clinically relevant measures such as HADS and EQ-5D-5L. These findings address a critical gap in understanding how symptoms of anxiety and depression may relate to PHE and HRQL in this demographic.

2. Conceptualization and evidence from theory

We used established frameworks for SDM and HRQL to guide our study conceptually, as described in the introduction. Specifically, we highlighted the theoretical linkage between SDM and patient engagement as a pathway to improved health outcomes. We acknowledge that further elaboration on the theoretical foundations may strengthen the manuscript, and we have revised the introduction to more clearly explain this in more detail.

3. Cross-sectional design and methodological choices

We carefully selected a cross-sectional design to assess relationships between SDM, PHE, and HRQL at a single point in time during clinic visits. This design was appropriate for our objectives and was feasible for engaging with emerging adults in mental health settings who may be hesitant to participate in research. While qualitative methods could provide deeper insights, our primary aim was to quantify relationships and establish preliminary associations. We have now clarified this rationale further in the revised manuscript in the methods section.

4. Research gap and literature review

The assertion that “not a single study has been conducted on predictors of SDM among emerging adults” stems from our published systematic review of SDM. While SDM has been studied in other populations, our manuscript explicitly highlights the scarcity of research addressing SDM predictors in emerging adults within primary care and mental health contexts. We recognize the need for a stronger framing of this gap and will enhance the introduction with additional references to underscore this point.

5. Psychometric properties of the ASK-MI instrument

The ASK-MI instrument’s high Cronbach’s alpha values reflect exceptional internal consistency but may suggest redundancy on some of the questionnaire items. We agree that this warrants further discussion and interpretation in the manuscript. We have also now cited additional publications that have used this instrument.

Manhas KP†, Olson K, Churchill K, Vohra S, Wasylak T. Implementation of a novel rehabilitation model of care across Alberta, Canada: A Focused Ethnography. BMJ Open Quality 2021 Mar 23;10(1):e001261. Doi:10.1136/bmjoq-2020-001261

Manhas KP†, Olson K, Churchill K, Miller J, Teare S, Vohra S, Wasylak T. Exploring patient-centredness, communication and shared decision-making under a new model of care: community rehabilitation in Canada. Health & Social Care in the Community 2021;30(3):1051-1063. https://doi.org/10.1111/hsc.13304

Manhas KP†, Olson K, Churchill K, Faris P, Vohra S, Wasylak T. Measuring Shared Decision-Making and Collaborative Goal Setting in Community Rehabilitation: A Focused Ethnography Using Cross-Sectional Surveys in Canada. BMJ Open 2020;10(8):e034745. doi:10.1136/bmjopen-2019-034745

Manhas KP†, Olson K, Churchill K, Vohra S, Wasylak T. Experiences of shared decision-making in community rehabilitation: a focused ethnography. BMC Health Services Research. 2020 April 19; 20(1):329: https://doi.org/10.1186/s12913-020-05223-4

6. Sample size and generalizability

While our sample size is modest, it aligns with typical feasibility constraints in clinical research involving emerging adults and outpatient settings. The observed lack of variability in SDM ratings is noted as a limitation in the discussion, and we remain transparent about the study's generalizability.

7. Interpretation of results

We acknowledge the importance of robust statistical interpretation and are prepared to refine and expand the discussion of our results to address this feedback. Specifically, we can provide greater depth in explaining the lack of variability in SDM ratings and the significance of the associations between anxiety, depression, and HRQL.

---

## [Decision Letter · Decision Letter 2]

1 Sep 2025

Dear Dr. Vohra,

Thank you for submitting your manuscript to PLOS ONE. After careful consideration, we feel that it has merit but does not fully meet PLOS ONE’s publication criteria as it currently stands. Therefore, we invite you to submit a revised version of the manuscript that addresses the points raised during the review process.

**Dear Respectable Authors**
**Based on feedback from reviewers, we make a decision regarding your manuscript.**
**Our decision is: Major revision**

We look forward to receiving your revised manuscript.

Kind regards,

Morteza Arab-Zozani, Ph. D.

Academic Editor

PLOS ONE

Journal Requirements:

1. "We note that there is identifying data in the Supporting Information file < SDM Emerging Adult Minimum Dataset July 23 2024.xlsx >. Due to the inclusion of these potentially identifying data, we have removed this file from your file inventory. Prior to sharing human research participant data, authors should consult with an ethics committee to ensure data are shared in accordance with participant consent and all applicable local laws.

- Name, initials, physical address

- Ages more specific than whole numbers

- Internet protocol (IP) address

- Specific dates (birth dates, death dates, examination dates, etc.)

- Contact information such as phone number or email address

- Location data

- ID numbers that seem specific (long numbers, include initials, titled “Hospital ID”) rather than random (small numbers in numerical order)

Please remove or anonymize all personal information, ensure that the data shared are in accordance with participant consent, and re-upload a fully anonymized data set. Please note that spreadsheet columns with personal information must be removed and not hidden as all hidden columns will appear in the published file.

"

2. Please include a copy of Table 1-3 which you refer to in your text on your paper.

Additional Editor Comments (if provided):

Reviewers' comments:

Reviewer's Responses to Questions

**Comments to the Author**

Reviewer #3: All comments have been addressed

Reviewer #4: All comments have been addressed

2. Is the manuscript technically sound, and do the data support the conclusions?

Reviewer #3: Yes

Reviewer #4: Partly

3. Has the statistical analysis been performed appropriately and rigorously?

Reviewer #3: N/A

Reviewer #4: I Don't Know

4. Have the authors made all data underlying the findings in their manuscript fully available?

Reviewer #3: Yes

Reviewer #4: Yes

5. Is the manuscript presented in an intelligible fashion and written in standard English?

Reviewer #3: Yes

Reviewer #4: Yes

Reviewer #3: (No Response)

Reviewer #4: Hello. Thank you for the opportunity to review this manuscript and answering questions of the previous reviewers.

I think this is a very important topic to study and needs further elucidation though I am concerned about the validity and significance of the results.

1) The study uses a dyad model in a situation where both the examiner and patient are aware of the results being studied. As mentioned in your limitations, it might introduce an amount of bias in the sample that it is difficult to read the significance of the results.

2) I think it would help to give more information about the ASK-MI questionnaire and what it measures including an example of the same to help clarify what it being measured. In the introduction, you mention the study being related to emerging adults with Anxiety/ Depression and their need/preference for SDM and/or their ability to participate in SDM. In the results however, if I'm understanding correctly, the ASK-MI questionnaire assesses their perception of SDM rather than their preference or necessarily ability but rather the HCPs involvement in engaging in SDM with each patient- which could also be the reason why there was no significant difference between the groups.

3) It makes sense that patients with current Anxiety/ Depression with a certain HADS cut off score would perceive that their quality of life is worse but would they perceive that the SDM is worse even if the HCPs provided the same amount of effort to it seems like the question being asked.

4) Overall, with these limitations, the validity of the findings in a curtailed sample size remains questionable. I think it would at the least, need a clarification of the above mentioned things.

**Do you want your identity to be public for this peer review?** For information about this choice, including consent withdrawal, please see our Privacy Policy

Reviewer #3: No

Reviewer #4: No

---

## [Author Response · Author response to Decision Letter 3]

16 Oct 2025

We thank the editors and reviewers for their careful review of our manuscript and for the constructive feedback. We appreciate the opportunity to revise further and clarify our work. Below, we provide our responses to each comment point-by-point.

Editor’s Comments

“We note that there is identifying data in the Supporting Information file < SDM Emerging Adult Minimum Dataset July 23 2024.xlsx >. Due to the inclusion of these potentially identifying data, we have removed this file from your file inventory… Please remove or anonymize all personal information, ensure that the data shared are in accordance with participant consent, and re-upload a fully anonymized data set.”

Authors’ Reply:

We thank the editor for this guidance. The dataset was approved for sharing by the University of Alberta's Research Ethics Board. To further reduce the risk of participant identification, we have grouped participant ages into ranges (e.g., 20–24, 25–29, etc.) and confirmed that all other variables (e.g., HADS scores, EQ5D, ASK-MI scores) are non-identifying. Patient IDs are sequential study IDs with no connection to hospital or system records. This de-identified dataset is now safe for public sharing in accordance with participant consent and our ethics protocol.

“Please include a copy of Table 1–3 which you refer to in your text on your paper.”

Authors’ Reply:

We will add Tables 1–3 to the revised manuscript in addition to uploading them separately to ensure that all text references correspond directly to the tables provided.

Reviewer #4

“The study uses a dyad model in a situation where both the examiner and patient are aware of the results being studied. As mentioned in your limitations, it might introduce an amount of bias in the sample that it is difficult to read the significance of the results.”

Authors’ Reply:

Thank you for this comment. To clarify, both participants and clinicians completed separate questionnaires and were not aware of each other’s responses. This design reduces the potential for direct response bias. However, we recognize that the dyadic model itself may still introduce subtle forms of bias. In the revised manuscript, we have strengthened the limitations section to acknowledge this and clarify how it may affect the interpretation of the findings.

“I think it would help to give more information about the ASK-MI questionnaire and what it measures including an example of the same to help clarify what it being measured.”

Authors’ Reply:

We thank the reviewer for this suggestion. In the revised manuscript, we have provided a more detailed description of the ASK-MI, including its purpose, structure, and administration. The ASK-MI includes parallel questionnaires for both participants and clinicians, which are completed independently. This design allows us to compare perceptions of shared decision-making without either party being aware of the other’s responses. We will also attach a supplementary file of each version of the document.

We have added the following text into the manuscript methods section (variables, shared decision-making) to clarify this issue.

“Neither the HCP nor the patient were aware of the results of the corresponding questionnaire.”

“In the introduction, you mention the study being related to emerging adults with Anxiety/ Depression and their need/preference for SDM and/or their ability to participate in SDM. In the results however, if I'm understanding correctly, the ASK-MI questionnaire assesses their perception of SDM rather than their preference or necessarily ability but rather the HCPs involvement in engaging in SDM with each patient- which could also be the reason why there was no significant difference between the groups.”

Authors’ Reply:

We appreciate this observation. We have clarified throughout the introduction and discussion that our study measured participants’ perceptions of HCP engagement in SDM, rather than their preference for or ability to engage in SDM. We have highlighted this distinction in our interpretation of the findings.

Reviewer #4

“It makes sense that patients with current Anxiety/ Depression with a certain HADS cut off score would perceive that their quality of life is worse, but would they perceive that the SDM is worse even if the HCPs provided the same amount of effort to it seems like the question being asked.”

Authors’ Reply:

Thank you for this thoughtful comment. Our initial hypothesis was that patients with higher anxiety and depression would be less likely to engage in SDM, even when providers made efforts to support it. Our findings, however, suggest that involvement in SDM may be more complex than patient symptom burden alone. It is possible that healthcare provider engagement plays a critical role in shaping patients’ perceptions of SDM, and that factors such as the amount of time spent with patients or whether they were first-time versus returning patients may also influence these perceptions. We have expanded the discussion to highlight these possibilities and to note that SDM may not be solely determined by patient characteristics, but also by contextual and relational factors.

Reviewer #4

“Overall, with these limitations, the validity of the findings in a curtailed sample size remains questionable. I think it would at the least, need a clarification of the above mentioned things.”

Authors’ Reply:

We agree that sample size is a limitation and will expand our discussion to explicitly acknowledge its impact on the generalizability and validity of the findings. However, the larger issues may be more related to low natural variability in SDM scores (mostly high ratings). The clinical settings we picked tended to have longer appointments with patients (often 30 mins to 1 hour) which may enable the perception of more SDM occurring. Preexisting awareness of the concept being studied also likely encouraged participants (clinicians) to “perform well”, leading to high scores. More work is needed in future research to limit these sources of bias to observe more natural variability on SDM being done.

Reviewer #3:

“All comments have been addressed.”

Authors’ Reply:

Thank you

---

## [Editor Report · Decision Letter 3]

28 Oct 2025

Predictors of shared decision-making among treatment-seeking emerging adults in primary care and community addiction and mental health settings: a cross-sectional study

PONE-D-23-23994R3

Dear Dr. Vohra,

We’re pleased to inform you that your manuscript has been judged scientifically suitable for publication and will be formally accepted for publication once it meets all outstanding technical requirements.

Kind regards,

Morteza Arab-Zozani, Ph. D.

Academic Editor

PLOS ONE
---

## [Editor Report · Acceptance letter]

PONE-D-23-23994R3

PLOS ONE

Dear Dr. Vohra,

I'm pleased to inform you that your manuscript has been deemed suitable for publication in PLOS ONE. Congratulations! Your manuscript is now being handed over to our production team.

Kind regards,

on behalf of

Dr. Morteza Arab-Zozani

Academic Editor

PLOS ONE